# Direct Memory Access-Based Data Storage for Long-Term Acquisition Using Wearables in an Energy-Efficient Manner

**DOI:** 10.3390/s24154982

**Published:** 2024-08-01

**Authors:** Cosmin C. Dobrescu, Iván González, David Carneros-Prado, Jesús Fontecha, Christopher Nugent

**Affiliations:** 1Departament of Information Technologies and System, University of Castilla-La Mancha, Paseo de la Universidad 4, 13071 Ciudad Real, Spain; ivan.gdiaz@uclm.es (I.G.); david.carneros@uclm.es (D.C.-P.); jesus.fontecha@uclm.es (J.F.); 2School of Computing, Ulster University, Belfast BT15 1ED, UK; cd.nugent@ulster.ac.uk

**Keywords:** long-term monitoring, continuous monitoring, ultra-low power data storage, embedded storage management, wearable devices, DMA controller

## Abstract

This study introduces a lightweight storage system for wearable devices, aiming to optimize energy efficiency in long-term and continuous monitoring applications. Utilizing Direct Memory Access and the Serial Peripheral Interface protocol, the system ensures efficient data transfer, significantly reduces energy consumption, and enhances the device autonomy. Data organization into Time Block Data (TBD) units, rather than files, significantly diminishes control overhead, facilitating the streamlined management of periodic data recordings in wearable devices. A comparative analysis revealed marked improvements in energy efficiency and write speed over existing file systems, validating the proposed system as an effective solution for boosting wearable device performance in health monitoring and various long-term data acquisition scenarios.

## 1. Introduction

Data storage is a critical component of wearable devices that directly affects their autonomy, data acquisition rate, accuracy, data persistence, and size constraints. Striking the right balance among these factors is essential for developing effective wearables capable of long-term and continuous monitoring, especially in health and well-being contexts [1,2]. Achieving this balance often involves trade-offs that can compromise data gathering, device battery life, or processing power.

Long-term and continuous monitoring through wearable devices is a crucial aspect in many human health and well-being contexts [1]. Depending on its use, a particular wearable device can implement a long-term strategy, but not continuous monitoring in some contexts [3,4]. In other cases, however, there is a need for continuous monitoring. For example, when focusing on monitoring vital signs, it is usually fundamental, as certain events manifest themselves punctually. These events must be pre-studied and wearable devices must be adapted so that they can be captured without loss of information [5,6,7,8,9,10]. The current availability of low-cost mass storage systems facilitates monitoring, thereby enabling broader data analyses. A large time series may reveal underlying patterns, and long-term data allow for retroactive reprocessing as new analysis techniques emerge, enhancing the quality of information extraction from the acquired data.

In long-term monitoring schemes, wearables typically employ three main approaches: direct transmission to a secondary device [7,11,12,13,14], local storage with periodic transmission [5,8,9,10,15,16,17,18,19], or on-device processing with the transmission of results [20,21]. Each method has its own advantages and limitations, particularly in terms of energy efficiency and data integrity. Regardless of the chosen approach, energy consumption remains a critical factor in all wearable designs.

Energy consumption remains a primary concern in wearable designs. This has been a constant challenge, leading to the development of different strategies to extend autonomy [22]. Although high-capacity batteries are available, they are often too large for wearable size limitations. Wearable devices must be aesthetically pleasing or at least unobtrusive [23,24], and optimizing energy consumption is crucial for both autonomy and user acceptance [25,26,27,28]. Various technical solutions have been explored to address these energy challenges.

The energy efficiency of wearables can be improved by various means, including the choice of communication protocols. Among the common interfaces, such as UART, I2C, and SPI, the Serial Peripheral Interface (SPI) protocol demonstrates the lowest power consumption, making it an excellent choice for wearable devices [29]. The use of Direct Memory Access (DMA) further enhances energy efficiency by significantly reducing power consumption in both the data acquisition and storage phases [20,30,31]. Combining SPI and DMA enables efficient data transfer between the microcontroller and storage devices such as Secure Digital (SD) cards, which are commonly used in wearable devices for mass storage. This approach allows the CPU to enter low-power sleep modes while data transfers occur independently, thereby further reducing the overall energy consumption. Although these hardware-level optimizations provide significant benefits, existing software solutions often abstract away from hardware specifics, potentially losing many of the advantages offered by the underlying hardware. A more effective approach is to develop a hybrid solution that leverages hardware capabilities through software design, even if this increases the complexity and reduces the flexibility.

Despite the existence of embedded file systems that can be executed in low-power embedded devices, these are generally reduced/simplified implementations of traditional file systems adapted for devices with limited resources. Most still maintain the same high-level logical structure of generic file systems (organized into directories and regular files). These types of systems are very CPU-dependent in their basic operations, not taking proper advantage of Direct Memory Access (DMA) features, energy-saving sleep modes, and are not intended for mass storage of time-series data. In other words, in the context of wearables where very specific functionalities are required, such as time-series data acquisition and storage, the generalization or use of a generic file system lightened in some form may have disadvantages compared to a dedicated data storage solution.

This work presents a novel approach to provide mass data storage in an energy-efficient manner that has been designed from the ground up. The solution enables long-term and continuous monitoring sessions and involves the combined use of (i) an SPI communication standard between the microcontroller and the SD card, (ii) a DMA to access the main memory independently of the CPU (avoiding cycles), and (iii) the use of sleep modes that are present in modern microcontrollers and allow for a drastic reduction in power consumption, while the DMA channels can perform concurrent read/write operations between the storage (SD card) and the main memory (RAM) of the wearable device.

The remainder of this paper is organized as follows. Section 2 presents the design of the proposed system by integrating both the software and hardware components. It covers the efficient organization of sensor data to minimize storage and maximize transmission efficiency, the alignment of data structures with communication protocols such as SPI for SD card interactions, and the introduction of a Time Block Data structure for control data management. Section 3 discusses the comparative analysis between the proposed system and the FatFs file system, including a detailed description of the experimental setup utilized for system evaluation. Section 4 analyzes the significance of these results, emphasizing the improvements in energy efficiency and data transfer rates. Finally, conclusions are presented in Section 5.

## 2. Materials and Methods

High-frequency sensor data acquisition inherently demands compromises in terms of battery life, data accuracy, and device size. It is essential to evaluate these interconnected characteristics during the design process to create devices that optimally balance storage capacity, data quality, and usability. Traditional file systems are not well suited for storing time-series data because of their limitations, such as those listed below. Extensive metadata management and hierarchical organization optimized for general storage creates overhead and slows down sequential access, which is crucial for time-series data. In wearable devices, where data loss might be tolerable in certain scenarios and vary depending on the application, these limitations can significantly impact performance and energy efficiency.

To address these challenges, this study proposes a lightweight storage system that is specifically designed for wearable devices. By minimizing the control data using innovative techniques, our system achieves significant reductions in memory usage, leading to improved performance and extended battery life. The file system has two goals:i.Optimizing energy efficiency;ii.Enabling data synchronization with other devices, with adjustable precision.

The proposed DMA-based storage system requires three key hardware features: a DMA controller, SPI interface, and interrupt support. These components are common on many modern low-power microcontrollers used in embedded systems. Implementing this approach on devices with these features can provide significant efficiency gains without requiring additional specialized hardware, thus minimizing the impact on device cost and complexity.

Optimizing energy usage in wearable devices involves leveraging the unique structure of sensor data (Section 2.1) and utilizing specialized hardware resources (Section 2.4). Thus, the integration of DMA and SPI protocols plays an important role in this strategy. DMA allows direct memory-to-peripheral data transfers, bypassing the CPU and minimizing data movement overhead. This collaborative interaction between DMA and SPI significantly reduces communication time and energy consumption within the device, making it ideal for battery-powered wearable applications.

The second goal is to provide data synchronization across multiple wearable devices, which is essential for accurate data analysis and insight generation. This requires highly accurate internal clock measurements for each wearable device involved. However, maintaining such high accuracy over time is difficult owing to time drift, a common issue where small variations in the clock’s internal components cause it to slowly lose sync. Implementing solutions that depend entirely on hardware for high-precision clock synchronization can be costly, both financially and in terms of energy consumption. This highlights the advantages of the alternative approach proposed in this study, which integrates periodic timestamp updates to maintain synchronization in a cost-effective manner. This approach offers the possibility of integrating periodic updates of timestamps (resynchronization updates) between data transfer operations (as an intrinsic part of them), thereby providing a cost-effective solution that matches the required time accuracy for most data analysis tasks.

### 2.1. Introduction to Sensor Data Packaging

Multilevel data organization improves adaptability across various applications. At the basic level, different raw sensor data are characterized by variable accuracy and quantization requirements. This means that, often, when transferring raw sensor data, bytes are not fully used (at the bit level); in other words, some bits are wasted and only used as padding to complete byte words (byte level), not containing sensor data information. This presents unique challenges, particularly in energy-efficient contexts where optimizing the data at a bit level can lead to significant power savings. Although bit-level adjustments require additional processing, they offer substantial benefits in terms of memory efficiency.

Consider the efficiency of storing data from a two-axis accelerometer with a 12-bit resolution per axis. Instead of using four bytes (32 bits) to store two 16-bit integers, a more efficient strategy combines 24 bits of actual data into three bytes. This is achieved by distributing the bits from both axes across the three bytes, optimizing memory usage without data loss. This method reduces the storage requirements from four bytes to three bytes, leveraging the available bits more efficiently.

Although the two-axis accelerometer example illustrates efficient bit-level data packing, similar principles are applied more broadly in our proposed system, encompassing various sensor types. Expanding on this example, our proposed system generalizes an efficient data packing approach to accommodate multiple sensor types, each contributing uniquely to the collective data architecture. The sensor data underwent a multi-layered transformation, as illustrated in Figure 1. From left to right, the raw sensor values were first adjusted at the bit level and organized into sensor samples.Each sensor sample represents multi-sensor readings from a wearable device at a specific acquisition time. Consecutive sensor samples are then combined into sensor data packages, which are a higher abstraction layer specifically designed for transferring a set of subsequent data samples, thus creating a larger data acquisition time window. Data packages include essential control information such as tokens for identification, a Cyclic Redundancy Check (CRC) for error detection, and an additional dummy byte (details explained in Section 2.4).

It is important to note that the choice of this transfer structure is not arbitrary; it directly mirrors the data transfer structures used by the SPI protocol with SD cards. As detailed in Section 2.2, these protocols and structures were adopted in our system to enhance the data processing and transfer efficiency. This approach not only optimizes memory usage but also ensures compatibility with existing infrastructures for SD card data transfer over SPI, facilitating implementation and scalability.

Thus, this approach to data packing is an integral part of our proposed system, ensuring that each sensor’s data are stored compactly and efficiently, thereby minimizing the memory usage. Importantly, the packaging process is designed to be customizable for each implementation and tailored to the specific requirements and sensor configurations of the application. This modular design allows for flexible adaptation to diverse operational requirements. The detailed management of these customizable packaging options is further explored in Section 2.3, which outlines how each application can develop its own version based on available sensors.

### 2.2. Data Structures and Transfer Protocols

To effectively organize data in the main memory, it is helpful to understand the low-level structure of the SD cards. These comprise a NAND flash memory and an embedded microcontroller (MCU), which is responsible for the management of memory operations. The embedded MCU facilitates data handling and communication with external devices through two primary protocols: Serial Peripheral Interface (SPI) and Secure Digital (SD) interface. SPI operates on a 1-bit bus, whereas the SD interface uses a 4-bit bus. SPI communication is generally simpler because it often omits the need for cyclic redundancy check (CRC) calculations on exchanged packages.

Interactions with SD cards over SPI are managed using specific commands [32,33], and establishing data transfer between devices requires setting up these commands for the read and write operations. The data were then exchanged using the data package structure, as depicted in Figure 2, which is almost identical to the structure used in the proposed system, as shown in Figure 1. These packages include a 1-byte start token, data block payload, and two CRC bytes, which are typically disregarded. Arranging data structures in the main memory as close as possible to the transfer packages helps reduce the processing required. The less complex and fewer control data a communication system has, the more efficient it becomes.

While SPI communication often omits CRC calculations for simplicity, the SD specification provides options for enhanced data integrity. According to the Physical Layer Simplified Specification [32], in Section 7.2.2 Bus Transfer Protection, CRC protection can be enabled using the CRC_ON_OFF command (CMD59). This feature is particularly valuable in environments susceptible to electromagnetic interference or when data integrity is paramount. It is important to note that when using SPI at moderate speeds, which are typical for wearable devices, and with a reasonably good hardware design, the decision to enable CRC calculation can depend on the fault tolerance requirements of each specific device. Although the SPI mode initializes with CRC disabled by default to optimize efficiency, implementing CRC can offer an additional layer of error detection when required for specific applications or environmental conditions.

Standard read/write operations typically involve a 512-byte payload size, which corresponds to the sector size of the card. This size is locked in during the initial power-up, with some cards offering no options for modification. During SPI read/write operations, the master MCU instructs the SD card regarding the number of data packages to be transferred. After each data package transfer, the card performs an internal verification and transmits success/error feedback via the MISO channel. This feedback informs the master MCU for the next data transfer decision. For a deeper insight into the transmission process, refer to Section 2.4.

### 2.3. SD Card Operations and Time Block Data Management

From the overall perspective of the proposed data storage system, Figure 3 shows the path from sensor data acquisition to the final storage on the SD card. Initially, the data acquired by the sensors are processed by the microcontroller, which organizes and temporarily stores them in the main memory until they are full. Subsequently, the DMA process is triggered by transferring the data to the SD memory card via the SPI connection. This scheme introduces Time Block Data (TBD), representing the intervals of data acquisition over a given time and with variable lengths.

A Time Block Data (TBD) structure consists of a header and a payload with a variable number of data packages (similar to those illustrated in Figure 1 when the data packaging procedure is presented). The structure is shown in Figure 4.

Each TBD points to the next one in the sequence, enabling flexible and targeted data access. This design optimizes sequential read/write operations while allowing efficient skipping of multiple TBD for the faster retrieval of specific information. The payload within each TBD holds data packages containing raw sensor data and their corresponding acquisition timestamps, based on their position in the series.

Detailing the organizational structure of the data, the TBD header is a control element. The header control information does not fully utilize the 512 bytes of a sector (payload size that is as common as the minimum transfer unit frequently used by SD cards). This leaves unused bytes within the header, allowing for more control variables/features to be added as required. These new variables/features are placed at the end of the TBD header to ensure backward compatibility. Table 1 defines in detail all variables at the header level of a TBD.

The TBD header or control sector is stored at the beginning of each TBD (first TBD sector). Given its importance, redundant information can be included within a sector by providing an additional data security layer. As the sector has the minimum transfer size, this does not add extra processing. For even greater fault tolerance, multiple sectors can be dedicated to storing copies of control information. Although this slightly increases the implementation complexity and incurs an extra energy cost, the desired level of fault tolerance can be tailored to specific design requirements. The proposed data storage system ensures that all the data acquisition processes begin with the creation of a new TBD.

This structure offers several advantages, including dynamic adaptation to evolving data acquisition requirements, easy timestamp incorporation, adjustments to data acquisition frequencies, sensor selection flexibility, and changes in sensor value resolutions.

Several factors influence the transition between the end of one TBD and the start of another. These factors include (i) updating the timestamp to prevent time drift, (ii) changing the frequency to adjust the data acquisition rate, and (iii) modifying the versioning to update the data packaging method. It is important to note that within a single TBD, the base frequency remains constant. However, different sensors can operate at various frequencies, provided that they are integer divisions of the TBD’s base frequency. This versioning change can involve alterations, such as changing data resolution, modifying sensors, or implementing a compression layer. Any changes to the base frequency or sensor configuration can only be implemented at the start of a new TBD to ensure data consistency within each block.

### 2.4. DMA Channels and Interrupt Service Routine

Once a particular multilevel structure has been detailed, using variable-length and linked Time Block Data (TBD), the focus shifts to data transfer between the microcontroller main memory and the non-volatile SD card memory through DMA and SPI. This subsection details the general DMA configuration, required interrupts, and communication timing.

The SPI protocol plays an important role because of its specific characteristics. It operates on a master–slave architecture, where the microcontroller acts as the master, controlling the data flow to and from the peripheral devices (slaves). The protocol uses dedicated lines for data transmission (MOSI) and reception (MISO) along a clock line (SCK) that synchronizes data exchange. The SPI module in the microcontroller includes hardware input and output registers that serve as temporary buffers for data before they are transmitted to or received from the peripheral device. These buffers are relatively small, holding sufficient data for immediate transfer needs but not for extensive data storage.

SPI buffer bits directly correspond to signals sent via MOSI and received via MISO lines. When data are written into the SPI transmit buffer by the DMA, each bit is sequentially converted into a signal that travels across the MOSI line to the slave. Similarly, the data bits received via MISO are stored in the receive buffer. The timing of the signal duration and the change in the physical channel are configured in the master and managed by the SCK signal.

These dedicated buffers are essential because they enable data transfer without CPU intervention. These are the key components that facilitate the effective use of DMA by providing temporary storage for data before they are transmitted or after they are received. Specifically, DMA accesses these buffers to transfer data from the main memory and converts them into physical signals.

Although minimal CPU intervention is still required to coordinate and oversee components at key stages for timing, error handling, and event responses, DMA overcomes the bottleneck of CPU-based data transfers. This hardware capability, executed through specific DMA channels, allows for Direct Memory Access and simultaneous data transfer activities without constant CPU involvement.

Coordination is achieved through Interrupt Service Routines (ISRs) that respond to specific events during the transfer process. ISRs ensure that the system operates reliably and maintains data integrity by responding to events, such as the completion of data transfers or the availability of new data.

The multilevel structure described in Section 2.1 combined with the SPI protocol, DMA system, and ISRs make up the essence of the proposed lightweight storage system that manages data acquisition. From the highest level of abstraction, data are prepared to ensure that the transmission is as unattended and efficient as possible.

Data management during write operations is achieved through various DMA channels, each with a specific role in the data transfer process. Figure 5 provides a graphical representation of the interactions and timing between the DMA channels and ISRs, as detailed in the following description of the data transfer cycle.

The Transmitter DMA (TxDMA) channel is primarily responsible for transferring data from the main memory of the microcontroller to the input registers of the SPI module. By directly moving data to these registers, TxDMA ensures that the SPI module has a continuous supply of data to transmit over the MOSI line, optimizing the transmission process.

The Receiver DMA (RxDMA) channel is activated towards the end of the data transmission process, as shown in Figure 5. It is responsible for capturing the response from the SD card. Once TxDMA completes the data transfer to the SPI, RxDMA prepares to receive the final byte or bytes from the SD card. These bytes indicate the success or failure of data transmission and are necessary to determine the next steps in data handling and management.

The final DMA channel (WaitDMA) is important for maintaining synchronization in the master–slave configuration of the SPI protocol. Although the primary data transfer concludes with TxDMA, the SD card requires time to process the received data and respond. During this interval, it is necessary to keep the clock signal (CLK) of the SD card active to maintain the communication channel. WaitDMA achieves this by sending dummy data (commonly 0xFF) over the MOSI line, which keeps the CLK signal oscillating until a response is received via MISO. Considering this complete sequence of operations, the DMA channels described (WaitDMA, RxDMA and TxDMA, as depicted in Figure 5) work in concert to ensure efficient and reliable data transfer within the proposed system, minimize CPU intervention, and enhance overall performance.

The proposed system utilizes three specific ISRs to manage data transfers and system responses: TxDMA Done, SPI Done, and uSD Ready. The ISRs and their interactions are illustrated in Figure 5. In addition, this type of embedded system employs several ISRs to manage data transfer and system responses. The ISR for the TxDMA Done is triggered when the TxDMA channel completes its task of transfering data from the main memory to the SPI input registers. This routine prepares the system for subsequent steps in the data transmission process and enables RxDMA to capture the last response byte from the SD card, thereby indicating whether the transfer was successful. TxDMA ensures that SPI module buffers that are emptied during transmission are continuously refilled with new data. This process maintains a constant flow of data to the SPI module. When no more data remain in the memory to transfer, TxDMA triggers an ISR that enables RxDMA to capture the final response bytes from the SD card.

In this sense, there is an ISR (named “SPI Done” in Figure 5) used to confirm that all the data present in the SPI transmission buffer are sent over the MOSI line. This ensures that the buffer is ready for the next set of data, maintaining a continuous and efficient data flow. This service routine is crucial for managing the state of SPI buffers and ensuring that they are ready for subsequent data transmission.

Furthermore, another ISR (named “uSD Ready” in Figure 5) is designed to detect changes in the MISO line, indicating that the SD card has processed the received data and is ready to send a response. This service routine is implemented directly on the MISO signal. Depending on the nature of the response, this ISR determines whether more data packages need to be sent or, conversely, whether all data have been successfully transmitted and received, guiding the next actions in the data transfer protocol. This mechanism ensures that the system can dynamically respond to the state of the SD card and effectively manage the data transfer cycle.

The data transfer cycle begins with the saveBuffer() routine in the firmware, initiating the writing process by enabling TxDMA and SPI peripheral. TxDMA then ends and activates RxDMA via TxDMA Done ISR. The latter is performed before the SPI peripheral finishes transmitting all data from its buffer. RxDMA handles responses received from the SD card. Once the SPI peripheral completes its transmission, the SPI Done ISR is triggered, enabling WaitDMA. This allows the SD card time to complete internal writing. When ready, the SD card signals, through the MISO line, triggering the uSD Ready ISR. This ISR checks whether more packages need to be sent or if the transmission cycle has been concluded.

During write operations, the SD card sends a data-response token after receiving each data block, as detailed in the Physical Layer Simplified Specification [32] in Section 7.2.4 Data Write. The SPI Done ISR, triggered upon completion of the SPI transmission, analyzes this response to determine if the data were successfully received by the SD card. This response includes a CRC validation check (if enabled) and a general Write Error indication. By verifying the success of each data block transfer within the SPI Done ISR, the system can immediately detect any transmission errors and take appropriate action, such as retrying the transfer and ensuring data integrity throughout the write process.

## 3. Results

A comparative analysis was performed to analyze the energy consumption and write speeds between a FatFs file system that is commonly used in wearable devices and the proposed storage system. This analysis quantifies the improvements in power usage and data transfer efficiency brought about by integrating Direct Memory Access (DMA) and the Serial Peripheral Interface (SPI) protocol. The results of this comparison highlight the effectiveness of the new system and provide insights into its enhanced functionality and efficiency in managing data transfers.

### 3.1. Experimental Setup

The microcontroller used was a PSoC 4 (CYBLE-222014-01) [34] designed specifically for low-power consumption using an ARM Cortex-M0 architecture. Figure 6 shows the configuration used to measure power consumption. A constant voltage power supply (3.3 volts and a 2-ohm shunt) was used. For accurate measurement at each point in time, a digital oscilloscope Analog Discovery 2 [35] on 500,000 samples/s was used.

The results were analyzed from two perspectives: the power consumption of the two systems and the time taken to write a fixed number of bytes. To analyze the power consumption, a capacitor was added to flatten and lengthen the large power consumption peaks so that they could be easily captured by the oscilloscope. Because the capacitor distorts the power consumption periods and thus the time taken to write, tests were performed without this capacitor to estimate the time of the write periods in the two scenarios.

The analysis was conducted from two perspectives: power consumption and time required to write a fixed number of bytes in both systems. To enhance the detection and quantification of power consumption peaks, a capacitor was included in the setup. This capacitor helps stabilize and extend the peaks, making them easier to capture with the oscilloscope. Although the capacitor introduces a minimal delay in the power consumption profile, this effect is insignificant and was accounted for in our analysis. Importantly, because the capacitor was used consistently across both tested systems, it allows for a relative comparison that focuses not on precise quantification of power consumption but rather on comparing the efficiency between the two systems. This method ensures that any potential distortions affect both the systems equally, thereby maintaining the integrity of the comparative analysis.

To obtain realistic results, two microSD cards were used, one with 4 GB and the other with 8 GB. The aim was to observe the behavior of the system on different cards with two different storage capacities, ruling out specific improvements due to the particularity of the card used. In both comparison cases, the same communication frequency was used for the SPI protocol, that is, 8000 kbps, with an oversampling of six (minimum possible). This limitation is imposed by the capability of ARM Cortex M0, which can operate only at 48 Mhz.

An oscilloscope function generator was used to compare the writing of the data from the RAM to the SD card at different frequencies. This generates pulses that are connected to a pin of the SoC, which will be in deep sleep mode depending on the case, and will proceed to send a total of 15,872 bytes, which is half of the main memory of CYBLE-222014-01. In the proposed system, 31 data packages with a default sector size of 512 bytes were sent. Using the oscilloscope for this task prevents the microcontroller from performing additional tasks, and simulates a situation in which the acquisition rate of the microcontroller is defined by a real-time clock (RTC).

To rule out behavior favorable to one system or the other, 20,000,000 samples were obtained in each test, equivalent to 40 s at an acquisition rate of 500,000 samples/s with 32-bit float values. Finally, a variable that can influence power consumption is the frequency at which the complete buffer (the main memory where the generated data samples are temporarily stored) is written to the SD card memory. Tests were conducted at writing frequencies of 2 Hz and 5 Hz.

### 3.2. Evaluation

The evaluation of the proposed storage system compares it with FatFs, a common embedded file system used in wearable devices and Internet of Things (IoT) applications. As described in the experimental setup in Section 3.1, the high-precision digital oscilloscope provides accurate digitization of the energy consumption patterns, enabling a detailed comparative analysis of both systems.

The evaluation was structured into two main parts to highlight specific performance metrics. The first part, presented in Section 3.2.1, examines the behavior of both systems during data transfer and write cycles and compares their operational patterns. This analysis provides insights into how each system handles these crucial operations.

Section 3.2.2 presents the second part of the evaluation. It presents quantitative measurements of the energy efficiency and transfer time, highlighting the performance differences between the proposed system and the FatFs. Through this structured approach, the evaluation aims to provide a clear comparison of the systems’ capabilities, focusing on the key performance metrics that are critical in embedded and IoT applications.

#### 3.2.1. Storage System Behavior during Transfer and Write Cycles

Figure 7 illustrates the behavior of the proposed system and FatFs using 4 GB and 8 GB SD cards at frequencies of 2 Hz and 5 Hz. The capacitor in this setup smoothens the peak energy usage, allowing for a clearer view of the consumption patterns. Owing to the precise pulse generation of the oscilloscope, all writing intervals were aligned and started simultaneously, thereby eliminating potential variations that could occur based on the number of consumption intervals.

Frequencies of 2 Hz and 5 Hz were selected experimentally to optimize the system analysis. Frequencies above 5 Hz led to an overlap in the FatFs write cycles, resulting in constant CPU activity. Even at 5 Hz, instances occur where the duration of a write cycle can triple, as shown in Figure 8, causing subsequent cycles to start late due to unfinished previous cycles. This behavior was accounted for in the analysis by counting effective writing intervals. In these cases, the microcontroller is programmed to initiate writing either upon an ISR trigger or after the completion of the preceding write operation.

#### 3.2.2. Energy Consumption and Transfer Time Comparisons

Figure 9 presents a magnified view of the $V_{sh}$ pattern during data transfer to the SD card for detailed analysis. Writing operations using the proposed storage system (DMA-based), as opposed to FatFs (CPU intensive), not only accelerate the transfer process, but also significantly reduce energy usage. This improved efficiency is evident in the varied consumption patterns observed among SD cards, attributable to the different management of NAND technologies across various cards, reflecting diverse approaches to handling data storage.

The efficiency variations can be further attributed to differences in microcontroller management functionalities, card capacities, and manufacturing distinctions across models. These factors highlight the impact of SD card technology on the overall efficiency of embedded storage systems.

The signals displayed on the oscilloscope corresponded to the voltage drop across the shunt resistor ($V_{sh}$) in millivolts (*mV*). This information is used to determine the current through the entire circuit and the power consumption of individual components using Ohm’s law, Equation (Equation 1), and thus the power consumption of the individual parts using Equations (Equation 2) and (Equation 3). Where $I_{ps}$ is the current of the entire circuit (ps = power supply), $V_{sh}$ is the shunt voltage measured by the oscilloscope, $R_{sh}$ is the 2-ohm shunt resistor, $P_{sys}$ is the power (*W*) consumed by the MCU and the SD card (sys = system), $V_{ps}$ is the supply voltage (3.3 V), and $P_{sh}$ is the power consumed by the shunt.
(1)$$I_{\mathrm{ps}}=\frac{V_{\mathrm{sh}}}{R_{\mathrm{sh}}},\;$$
(2)$$P_{\mathrm{sys}}=I_{\mathrm{ps}}\times \left(V_{\mathrm{ps}}-V_{\mathrm{sh}}\right),\;$$
(3)$$P_{\mathrm{sh}}=\frac{V_{\mathrm{sh}}^{2}}{R_{\mathrm{sh}}},$$

The analysis involved 20 M samples, equivalent to 40 s, with 80 transfer cycles at 2 Hz and 200 transfer cycles at 5 Hz. The comparison between the two equivalent systems should not be confined solely to power consumption (W) as this might overlook significant variations in the writing periods, potentially biasing the analysis. Therefore, it is essential to delineate and specifically analyze active writing periods in the consumption signal for accurate assessment. This segmentation enables the focus on phases where consumption is most pertinent and distinctly defines the duration and intensity of data transfer periods.

Figure 10 illustrates that it is relatively easy to delimit the writing periods based on the consumption signal. To obtain the start and end points of all intervals, the signal is first smoothed to prevent points in a small region from having unexpected peaks that could distort the delimitation. Using the smoothed signal, all cuts were detected at an amplitude of 14 mV, which was selected as a practical threshold. This value of 14 mV was empirically determined by observing that it consistently represents a transition to a state of significant activity, effectively marking the commencement and conclusion of writing periods while avoiding false positives from minor signal fluctuations.

The 14 mV threshold (cutOffPoints), shown in Figure 10, will serve as a basis for refining the search for the writing cycle. With this initial approximation, the total number of points obtained is divided into odd and even points, with the odd points defining the start and even points defining the end of the cycles. To refine the start of the cycle, odd values are taken as reference points, and the signal is moved backward to reach a local minimum. Similarly, with the even values, the signal is advanced until a local minimum is reached. For this purpose, simply take a window of two values and scroll through the signal until the new one is no greater than the previous one. Following this procedure, the initial and final values that delimited the writing cycle were obtained.

To determine the power consumed by the system ($P_{sys}$), the power consumption is subtracted from the shunt resistance, and the energy consumed can be determined by the duration of the transfer cycles (time). To calculate the energy consumed during a specific transfer cycle, Equations (Equation 4) and (Equation 5) are used to determine the value of $E_{sys}$.
(4)$$E_{\mathrm{sys}}=P_{\mathrm{sys}}\times t,\;$$
(5)$$t=\frac{\mathrm{start}-\mathrm{end}}{\mathrm{sample}\;\mathrm{rate}},$$

Figure 11 compares the two storage systems based on the average energy consumption for writing 15,872 bytes. The DMA-based proposed system significantly reduces power consumption in all scenarios. At 2 Hz, power consumption is reduced by up to 3.4 times for the 4 GB card and 7.8 times for the 8 GB card. At 5 Hz, the difference increases further, with reductions of 3.7 times for the 4 GB card and up to 10.6 times for the 8 GB card.

The observed improvements in energy consumption stem from two factors: reduced instantaneous power draw for equivalent actions and faster write times. Figure 12 shows the average write time of 15,872 bytes from RAM to the SD card. The proposed DMA-based system is ×2.3 times faster at 2 Hz and ×2.2 times faster at 5 HZ in the case of the 4 GB card. On the 8 GB card, the DMA was ×3.9 times faster at 2 Hz and ×7.8 times faster at 5 Hz.

The improvements in the energy consumption stem from the reduced instantaneous power draw for equivalent actions and faster write times. Figure 12 shows the average write time of 15,872 bytes from RAM to the SD card. The proposed DMA-based system is 2.3 times faster at 2 Hz and 2.2 times faster at 5 Hz for the 4 GB card. For the 8 GB card, the DMA system was 3.9 times faster at 2 Hz and 7.8 times faster at 5 Hz.

## 4. Discussion

The proposed system leverages a common DMA architecture to offer a widely adaptable and efficient embedded data storage solution across various platforms. Performance improvements of up to 10.6 times compared to the FatFs file system were achieved. These gains result from (i) reduced data transmission time and (ii) the use of DMA to manage SD card transfers instead of the CPU. Offloading the DMA allows the CPU to enter low-power mode, as this module is more efficient for this task. Furthermore, the CPU becomes available for other processes during writing, which is advantageous for applications requiring continuous SD card operations alongside tasks such as data acquisition or other communications.

The system was specifically designed for wearable devices that periodically capture data by using sensors. This optimization was achieved by encoding the data capture time using a timestamp and its location in a temporal sequence. The key strategy is to use predefined capture intervals, thereby eliminating the need to store additional time information. This approach allows for flexibility in handling different capture frequencies, as long as a known a priori pattern is followed to decode the information.

Although this methodology may restrict its application to broader contexts, it offers sufficient versatility to adapt to various situations. By specifically modifying the header and adding control elements, it was possible to record the timestamp for each acquisition period. Thus, the system can be adjusted to maximize its efficiency based on the particular needs of each use case. Proper data interpretation is subject to information contained in the Time Block Data header. Any erroneous alterations to this header may hinder the data extraction. Depending on the specific context, security replicas can be generated in different memory areas to prevent data losses.

Traditional file systems such as FatFs can lead to inconsistent performances when used with SD cards for time-series data storage. More targeted solutions that optimize data structures and control procedures for this application improve performance and energy efficiency compared to general purpose file systems. In the proposed design, both energy consumption and write time were observed to be more consistent across different SD cards, reducing the variability introduced by the storage medium. With the 8 GB SD card, the DMA-based method consumes less energy and requires less time than FatFs. The extra consumption, compared to the 4 GB card and 5 Hz write, is due to the increased overlap between write intervals. The use of a generic file system causes many control instructions to be performed with the CPU, which slows down the writing. To maximize the transfer rates, it is recommended to increase the operating frequency of the SPI protocol within the limits of the chosen microcontroller. As an example, at 8000 kbps, the DMA system achieves an average write time of 30.4 ms for 15,872-byte packages. However, the proposed system is not designed for extremely high sustained write speeds because overlapping write periods can lead to data synchronization loss. Therefore, it is essential to evaluate the specific requirements for each case before implementing this storage system.

The SD card longevity in wearable devices depends on factors such as temperature, technology (SLC, MLC, and their write cycles), and capacity. Our system, which is designed for sequential time-series data, supports efficient cyclic writing. This allows new Time Block Data (TBD) writes to start at incrementally higher sectors after data deletion, evenly distributing wear across all sectors of the storage medium.

Future work will focus on further refining the system to accommodate a wider range of data acquisition scenarios and explore the integration of more complex data analysis techniques directly on wearable devices. This will not only expand the applicability of the proposed system but also pave the way for more autonomous and intelligent wearable solutions. The ongoing evolution of this storage system holds promise for transforming the wearable technology landscape, making long-term and continuous monitoring more feasible and effective for a broad spectrum of applications.

## 5. Conclusions

This study introduces a DMA and SPI protocol-based storage system for time-series data. It is primarily aimed at enhancing the energy efficiency of data-logging applications, notably within wearable technology for long-term and continuous monitoring. By replacing traditional file structures with Time Block Data units, the system minimizes control overhead by focusing only on the periodic recording of data. This approach significantly simplifies data management in applications such as long-term monitoring, where energy conservation is essential, owing to the limited energy resources of the wearable devices.

The proposed storage system provides an integrated solution that includes (i) exploiting the specific characteristics of time-series data, (ii) organizing these data into structures that facilitate efficient processing, and (iii) efficient use of hardware resources.

Testing with SD cards of various capacities revealed that the maximum energy efficiency improvement achieved was 10.6 times compared to traditional storage methods. This efficiency enables the CPU to enter low-power modes more frequently or to remain in these modes longer while simultaneously handling other tasks more effectively because of the reduced demand on its processing resources.

## Figures and Tables

**Figure 1 sensors-24-04982-f001:**
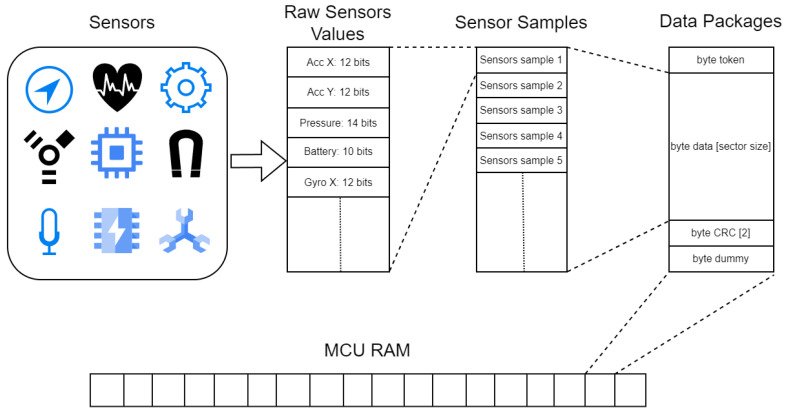
Multi-layered packaging for sensor data in main memory.

**Figure 2 sensors-24-04982-f002:**
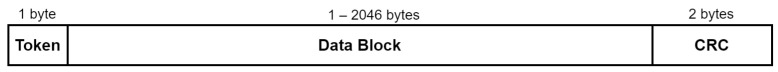
Minimal data transfer unit for SD card interactions via SPI commands, showing a starting token, a variable-length data block, and a terminating CRC.

**Figure 3 sensors-24-04982-f003:**
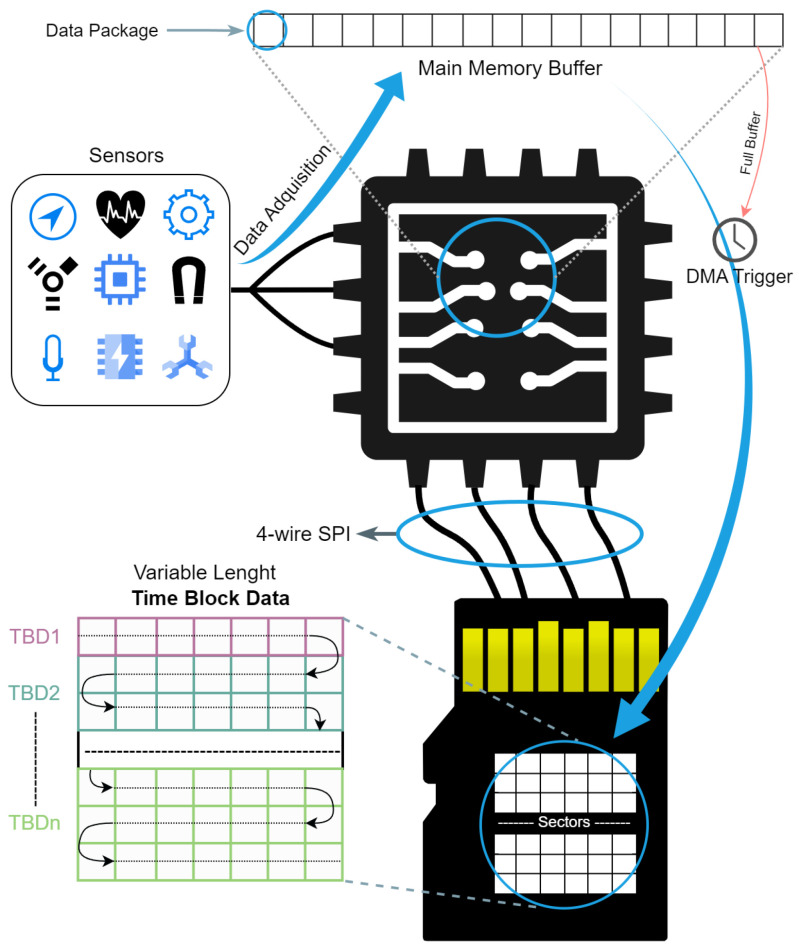
Data flow from sensor acquisition to SD card memory storage, detailing the microcontroller’s role, RAM buffer staging, and DMA-triggered SPI transfer, with focus on the variable-length Time Block Data (TBD) organization.

**Figure 4 sensors-24-04982-f004:**
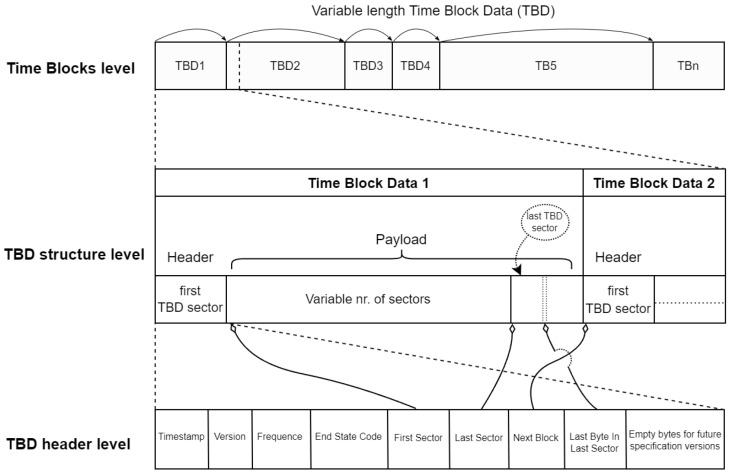
Three-level view of the Time Block Data (TBD) architecture: the Time Block level illustrates variable-length TBD and their links, structure level shows the internal composition with header and payload, and header level presents control information.

**Figure 5 sensors-24-04982-f005:**
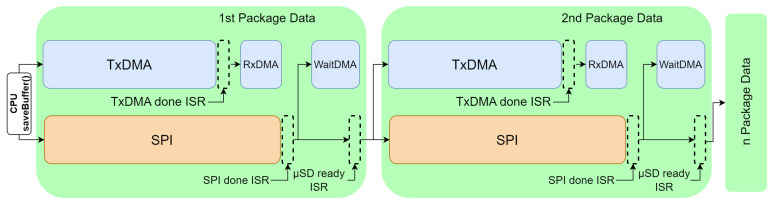
Behavior of DMAs, ISRs and the SPI module in the system-on-chip (SoC).

**Figure 6 sensors-24-04982-f006:**
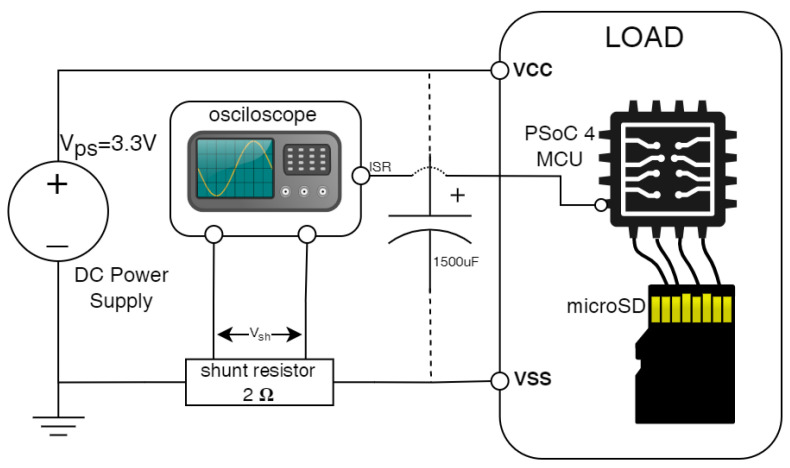
The power measurement setup utilized a 2-ohm shunt, a 3.3-volt power source, and an oscilloscope to obtain digital samples.

**Figure 7 sensors-24-04982-f007:**
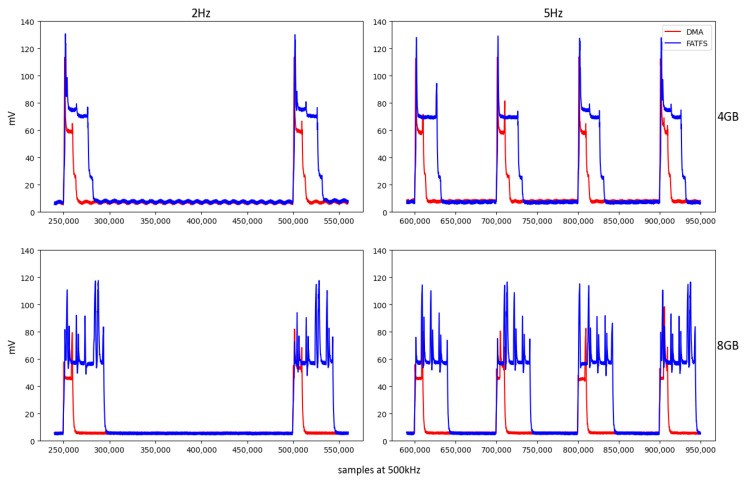
Evolution over time of the voltage drop across the shunt resistor ($V_{sh}$) during a write of 15,872 bytes. Variables include memory card capacity (4 GB vs. 8 GB) and write frequency (2 Hz vs. 5 Hz). In red, $V_{sh}$ is using the proposed storage system with DMA; and in blue color, $V_{sh}$ is using FatFs. The periods (and transfer peaks) can be easily distinguished in both 2 and 5 hertz frequency settings.

**Figure 8 sensors-24-04982-f008:**
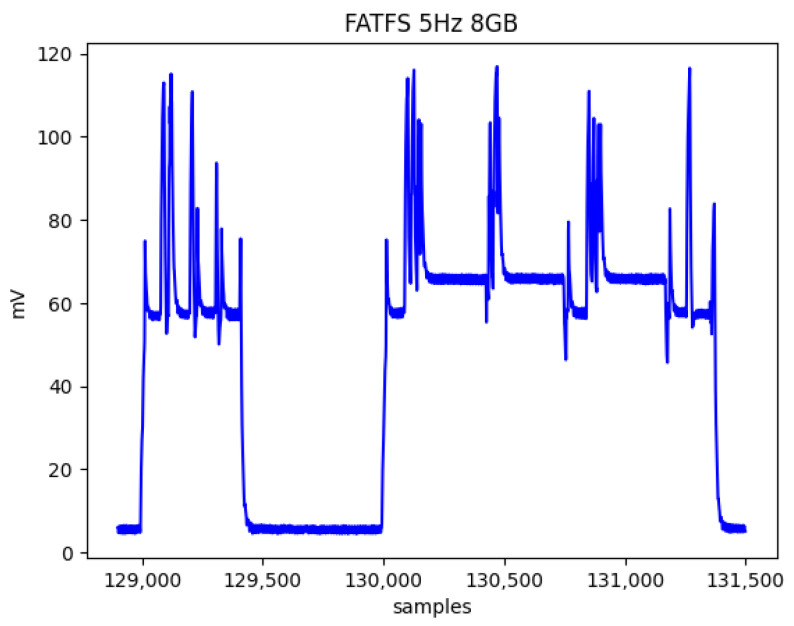
Three overlapping writing periods causing time synchronisation to be lost which leads to constant and continuous CPU activity.

**Figure 9 sensors-24-04982-f009:**
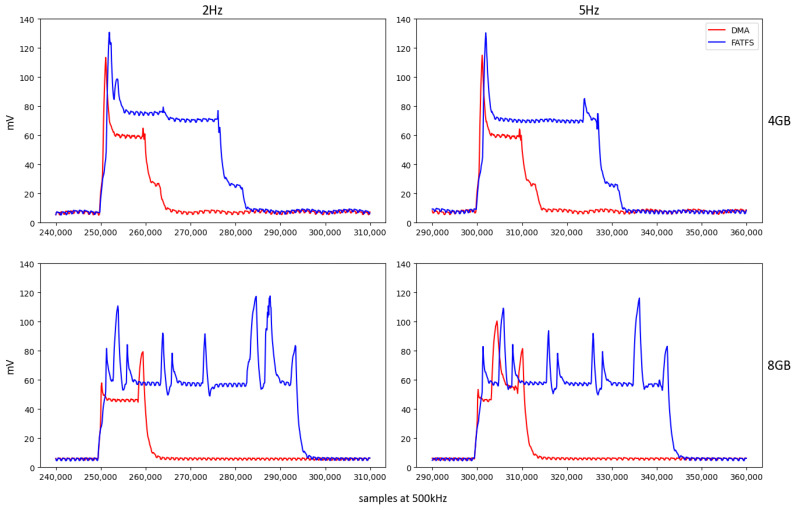
Zoomed-in view of $V_{sh}$ time evolution pattern for writing 15.872 bytes in each setting.

**Figure 10 sensors-24-04982-f010:**
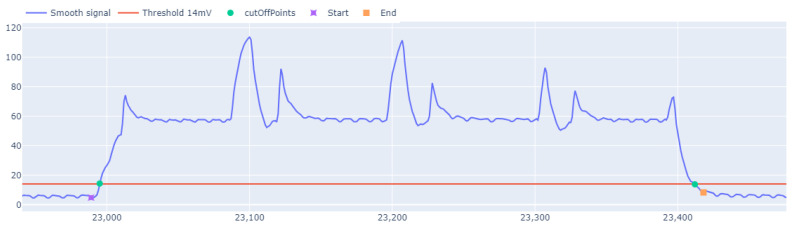
Delimitation of a transfer cycles based on the threshold. Cut-off points are used as a reference and they help to establish the start and end of the interval by going down the slope.

**Figure 11 sensors-24-04982-f011:**
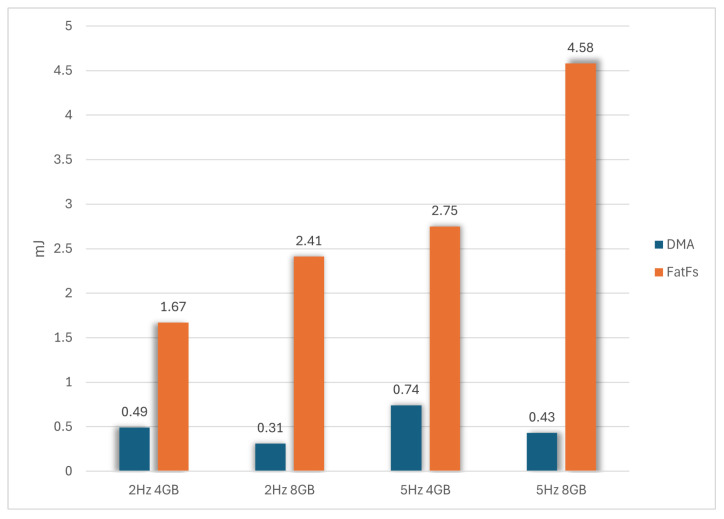
Average energy consumption after writing 15,872 bytes.

**Figure 12 sensors-24-04982-f012:**
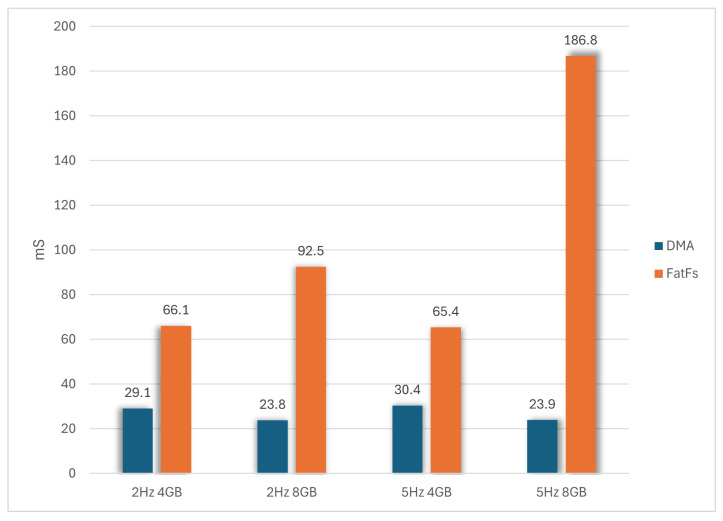
Average time required for writing 15,872 bytes.

**Table 1 sensors-24-04982-t001:** Variables defined at the header level of a TBD.

Variable	Description
Timestamp	Stores a time reference, working in conjunction with each data’s position.
Version	Indicates the data packaging method within the TBD.
Frequency	Specifies the data acquisition frequency within the TBD, enabling accurate timing for each stored data sample.
End State Code (ESC)	A control variable indicating the TBD’s state, requiring three values: ESC = 1 indicates the presence of a subsequent block ESC = 2 indicates writing in progress ESC = 3 indicates the time block is correctly closed, and there is no subsequent block
First Sector	Facilitates programming logic for sector reading and navigating through different TBD.
Last Sector	Indicates the final sector of a TBD. In the case of ESC = 2, the value should be 0 × FF to signal an error in closing TBD.
Next Block	Simplifies data access logic programming and shows the start of the next TBD.
Last Byte in Last Sector	Indicates the position of the last valid byte of data in the last sector of a TBD, used because data acquisition can stop at any time and the sector may not be fully filled.

## Data Availability

The measurements of electrical consumption, which were collected and analyzed for this study, will be available at https://zenodo.org/doi/10.5281/zenodo.13143215 (accessed on 29 July 2024).

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
