# Peer review of "Direct Memory Access-Based Data Storage for Long-Term Acquisition Using Wearables in an Energy-Efficient Manner"

_sensors, 2024, doi:10.3390/s24154982_

Round 1

Reviewer 1 Report

Comments and Suggestions for Authors

This study presents a DMA and SPI-based storage system that significantly enhances energy efficiency and write speeds in wearable devices by organizing data into Time Block Data units. By reducing control overhead, it enables efficient long-term and continuous monitoring.

Here are the questions and suggestions for the manuscript

1.The content from lines 347-352 is identical to that of lines 353-358.

2. How does the system address variations in data acquisition frequencies, and what mechanisms to adjust the frequency settings within TBD units?

3.Can the authors provide a detailed assumption of energy consumption for each component in the data transfer process (e.g., microcontroller, DMA controller, SPI bus, SD card)?

Reviewer 2 Report

Comments and Suggestions for Authors

This paper introduced a novel DMA-based data storage system for wearable devices aimed at enhancing energy efficiency during long-term and continuous monitoring. It leveraged Direct Memory Access and the Serial Peripheral Interface protocol to facilitate efficient data transfer, significantly reducing energy consumption and improving device autonomy. The system organized data into Time Block Data (TBD) units instead of traditional file systems, thereby minimizing control overhead and optimizing storage management for periodic data recordings. A comparative analysis demonstrates substantial improvements in energy efficiency and write speed over existing file systems, validating the proposed system as a superior solution for wearable device performance in health monitoring and various long-term data acquisition scenarios. The manuscript was well organized and constructed. The test data were depth analyzed and fully discussed. Therefore, the manuscript is recommended for publication in Sensors with minor revisions. Some specific comments are listed below.

(1) How did the proposed system handle data corruption or loss during DMA transfers, and what specific error correction mechanisms are in place?

(2) What specific hardware requirements are needed for implementing this DMA-based storage system in existing wearable devices, and how does this impact device cost and design complexity?

(3) The detailed benchmarks comparing the energy consumption and write speeds of the proposed system with other leading storage solutions across different types of wearable devices should be provided in the revised manuscript.

(4) What is the impact of the proposed system on the overall longevity and reliability of wearable devices, considering factors such as wear and tear on the SD card due to frequent read/write operations?

Comments on the Quality of English Language

Language is fine.

Reviewer 3 Report

Comments and Suggestions for Authors

The manuscript is too lengthy to be read. Thus, at this stage, the reviewer cannot make a judgement on the content. The authors are suggested to read some good papers published in top journals (such as Nature and Science) and re-write the manuscript. Specifically:

1. Introduction is composed of 10 paragraphs. I cannot catch the significance of this work from this long context.

2. I never see a "State of the Art" section (11 paragraphs) in a journal paper. This should be merged into Introduction. The significance of this work should be expressed in about 4 paragraphs. 

3. Materials and Methods section is also lengthy. Please refer to some Nature/Science papers. 

Publishing a paper is for communication. If it is un-readable, how could we communicate?

Comments on the Quality of English Language

No.

Round 2

Reviewer 3 Report

Comments and Suggestions for Authors

On page 4, the footnote of "1 Cyclic Redundancy Check" is not necessary because there is a abbreviation list at the end of the manuscript.

Comments on the Quality of English Language

The expression is suggested to be further simplified. 

Author Response

Thank you for your comments. We agree with your suggestions. We have addressed them as follows:

  1. We have removed the footnote for "Cyclic Redundancy Check" on page 4, as this abbreviation is already included in the list at the end of the manuscript.
  2. We have made minor revisions throughout the manuscript to improve clarity and readability, simplifying some expressions as suggested.

We appreciate your feedback, which has helped us improve the quality of our paper.